# Detecting the ultra low dimensionality of real networks

Pedro Almagro [1], Marián Boguñá [2,3] & M. Ángeles Serrano [2,3,4] ✉

Reducing dimension redundancy to find simplifying patterns in high-dimensional datasets and complex networks has become a major endeavor in many scientific fields. However, detecting the dimensionality of their latent space is challenging but necessary to generate efficient embeddings to be used in a multitude of downstream tasks. Here, we propose a method to infer the dimensionality of networks without the need for any a priori spatial embedding. Due to the ability of hyperbolic geometry to capture the complex connectivity of real networks, we detect ultra low dimensionality far below values reported using other approaches. We applied our method to real networks from different domains and found unexpected regularities, including: tissue-specific biomolecular networks being extremely low dimensional; brain connectomes being close to the three dimensions of their anatomical embedding; and social networks and the Internet requiring slightly higher dimensionality. Beyond paving the way towards an ultra efficient dimensional reduction, our findings help address fundamental issues that hinge on dimensionality, such as universality in critical behavior.

The problem of dimensions—identifying the dimensionality of the relevant space associated with a given phenomenon—recurs across disciplines in the natural sciences. In statistical physics, the number of spatial dimensions is one of the few factors that determines the critical properties and the universality class of extended systems where events at multiple length scales make relevant contributions. In the string theory framework for particle physics and quantum gravity, extra invisible dimensions lie beyond the three that we observe in ordinary space. However, those additional dimensions cannot be reached either because they could all be curled up into tightly packed manifolds or because, even though some of them could be large, events that we can experience are locked onto some subset of dimensions. This last idea also emerges in a completely different framework within computer science and network science: complex data and interactions only populate a small subspace of their original high-dimensional space. This blessing of dimensionality[1,2] —recast as a curse of dimensionality[3] in other contexts where the sparsity of data may be problematic—is

sustained by phenomena of measure concentration as dimension increases[4–6], which can greatly help mathematical analysis of real systems[7]. Assisted by these effects, reducing dimension redundancy to find simplifying patterns in high-dimensional datasets and graphs has become a major endeavor.

In the field of computer science, a variety of data-driven techniques have been proposed to facilitate this task[8–12]. These techniques are most often based on some definition of similarity distance between the elements in the dataset[13–15] and involve the construction of a similarity graph that is mapped onto a latent low-dimensional space, typically Euclidean, where connected nodes are kept close to each other[16–18]. However, the intrinsic geometry of complex datasets and graphs is not obvious, and defining similarity distances in agreement with their relational and connectivity structure is challenging. In addition, the graph embedding techniques employed often assume a latent space with a predetermined number of dimensions or implement heuristic techniques to find a suitable value for the embedding

[1]Departamento de Ciencias de la Computación e Inteligencia Artificial, Universidad de Sevilla, Sevilla, Spain. [2]Departament de Física de la Matèria Condensada, Universitat de Barcelona, Martí i Franquès 1, 08028 Barcelona, Spain. [3]Universitat de Barcelona Institute of Complex Systems (UBICS), Universitat de Barcelona, Barcelona, Spain. [4]Institució Catalana de Recerca i Estudis Avaçats (ICREA), Pg. Lluís Companys 23, 08010 Barcelona, Spain. ✉e-mail: marian.serrano@ub.edu

dimension, for instance by evaluating the quality of embeddings across different dimensions[12,19]. Moreover, different embedding models applied to the same network may lead to different values of the selected embedding dimension. Independent principled methods are thus required to find the intrinsic geometry and dimension of data with complex structure.

In the framework of network science, the characterization of dimensionality is linked to the definition of distance. The fractal dimension of a network has been defined on the basis of scaling properties under a length scale transformation using similarity distances defined in terms of topological shortest paths[20–28]. The same procedure has also been applied to networks using explicit geometry—for instance, geography—as the reservoir of distances, with the result that networks characterized by a wide distribution of link lengths in Euclidean space have a fractal dimensionality higher than that of the explicit space[29]. Alternatively, the dimensionality of such spatial networks has also been measured as a correlation dimension computed from time series that describe network trajectories of random walkers[30].

Here, we introduce a method to infer the dimensionality of the latent hyperbolic space underlying the connectivity of a complex network without the need for any a priori spatial embedding. We also avoid shortest path distances since they are strongly affected by the small-world property and do not provide a broad range of distance values. Within the context of network geometry[31], our approach is model-driven and assumes that real networks are well described by the geometric soft configuration model in $D$ dimensions, the $\mathbb{S}^D/\mathbb{H}^{D+1}$ model, which is a multidimensional generalization of the $\mathbb{S}^1$ model[32] and its $\mathbb{H}^2$ purely geometric formulation in hyperbolic space[33]. The $\mathbb{S}^1/\mathbb{H}^2$ model is based on fundamental principles to describe the observed connectivity of real unweighted and undirected networks. The model assumes a one-dimensional similarity space plus a popularity dimension from which hyperbolic geometry emerges as the geometry that naturally embodies the hierarchical architecture of networks. In this way, the model explains many typical features of real networks including sparsity, the small-world property, heterogeneous degree distributions, and high levels of clustering[34]. Furthermore, statistical inference techniques allow us to obtain maps of real networks in the hyperbolic plane that are congruent with the model[35]. Beyond visualization, these representations have been used in a multitude of tasks, including efficient navigation[36–38]; the detection of patterns such as self-similarity[32,37,39,40] and communities of strongly interacting nodes[41,42]; and the implementation of a renormalization procedure that brings to light hidden symmetries in the multiscale nature of complex networks[37,40] and enables scaled-down and scaled-up network replicas[43].

The geometric soft configuration model is able to do that while being a maximal entropy model, meaning that it makes the minimum number of assumptions to explain observations given the constraints (degree distribution and level of clustering). Hence, it is the most parsimonious option providing the simplest explanation for the complex topology of real networks[34]. As we show below, the multidimensional model can produce different graph structures—while preserving sparsity and the small-world property—by changing the degree distribution, parameter $\beta$ that controls clustering, and the dimension $D$ that controls the chordless cycles spectrum. This dependency of the densities of chordless cycles of different lengths on the dimensionality of the model is the feature that we exploit to implement our dimensionality detection methodology.

## Results

### Statistics of cycles and their relation with dimensionality

If networks are metrical and related to a latent space in such a way that connections between nodes are more likely the closer the nodes are in that space, then differences in the structure of the space due to

changes in its dimensionality should be naturally reflected in the topology of the networks. This suggests that measuring the intrinsic dimensionality of a complex network should be possible by computing profiles of structural properties that are expected to be sensitive to dimensionality. Synthetic surrogates produced with the generalization of the $\mathbb{S}^1/\mathbb{H}^2$ model to $D$ similarity dimensions, the $\mathbb{S}^D/\mathbb{H}^{D+1}$ model[32,34], can then be used to assess the statistics obtained. In what follows, we prove that the frequencies in the graph of chordless cycles of different lengths are just such key structural properties. A chordless cycle is defined as a closed path in the graph without a cycle chord, meaning that all edges between the nodes of the cycle belong to the edge set defining it. Persistent homology of complex networks[44–46] —a recently developed computational method that quantifies the stability of topological features (typically holes measured as voids bounded by simplices) in the simplicial complexes of a sequence of successive approximations of the original dataset—also focuses on the statistics of a particular type of cycles to obtain geometrical information. In contrast to persistent homology, in our work we compute local edge cycles in the original network and thus we are not restricted to simplicial complexes. More importantly, persistent homology does not shed any light on the manifold structure beyond its homology or on its relation with network connectivity and, even if the qualitative geometric structure of the data is detectable, the precise dimension is highly sensitive to noise and difficult to estimate using this technique, especially in medium-sized to large networks.

In the $\mathbb{S}^D$ model, a node $i$ is assigned two hidden variables: a hidden degree $\kappa_i$, quantifying its popularity or importance, and a position in a similarity space, represented as a $D$-dimensional sphere, denoted by $\mathbf{v}_i$. The probability of connection between any pair of nodes $i$ and $j$ takes the form of a gravity law, and its magnitude increases with their combined popularities and decreases as their similarity distance increases, so that more popular and similar nodes are more likely to be connected:

$$p_{ij} = \frac{1}{1+\chi_{ij}^\beta} \quad \text{with} \quad \chi_{ij} = \frac{R\Delta\theta_{ij}}{(\mu\kappa_i\kappa_j)^{1/D}}. \tag{1}$$

The hidden variable $\kappa_i$ of node $i$ is called its "hidden degree" because it coincides with the expected degree of node $i$ in the ensemble of graphs produced by the model. Thus, the degree distribution of these graphs is strongly related to the distribution of hidden degrees, $\rho(\kappa)$, which is arbitrary. To model the degree heterogeneity observed in real networks, here we chose $\rho(\kappa)$ to be power-law distributed, that is, $\rho(\kappa) \propto \kappa^{-\gamma}$ with $\gamma > 2$. $\Delta\theta_{ij}$ is the angular distance between the two vectors, $\mathbf{v}_i$ and $\mathbf{v}_j$, positioning nodes $i$ and $j$ in the $D$-dimensional similarity sphere of radius $R = \left[\frac{N}{2\pi^{\frac{D+1}{2}}}\Gamma\left(\frac{D+1}{2}\right)\right]^{\frac{1}{D}}$, where $N$ is the number of nodes in the graph so that, without loss of generality, the density of nodes on the $D$-sphere is set to 1. The parameter $\beta$ (or inverse temperature) calibrates the coupling of the network with the underlying metric space and controls the level of clustering in the topology (cycles of length three in the network induced by transitive relations) as a reflection of the triangle inequality in the latent geometry. Finally, the parameter $\mu$ controls the average degree of the network. The Fermi-Dirac form of the connection probability in Eq. (1) is the only possible choice that defines maximally random ensembles of geometric graphs which are simultaneously sparse, heterogeneous, clustered, small-worlds (if $\gamma < 3$ or $\beta < 2D$), and degree–degree uncorrelated[34]. In its isomorphic purely geometric formulation, the $\mathbb{H}^{D+1}$ model[33,47,48], the popularity and similarity dimensions are combined into a single distance in $(D+1)$ hyperbolic space.

We next prove that cycles of different length and dimensionality are intertwined in $\mathbb{S}^D$ networks in a non-trivial way, which ultimately enables us to determine $D$. In particular, clustering poses an upper limit on the dimension of the similarity space. As noted in

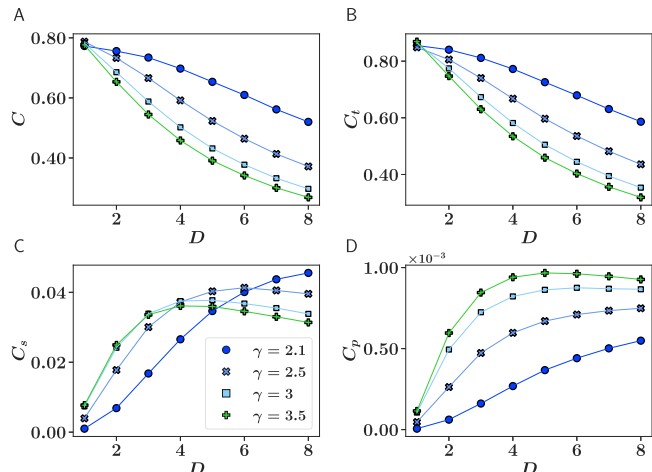

**Fig. 1 | Maximum clustering as a function of the dimension.** Maximum value of node clustering (**A**), and mean density of edge triangles (**B**), squares (**C**), and pentagons (**D**), as a function of the dimension $D$ and power law exponent $\gamma$ in synthetic networks generated by the $\mathbb{S}^D$ model. Values correspond to averages over 100 network realizations of size $N = 1000$ and $\beta = \infty$. The error bars are smaller than symbol sizes. The value of $\mu$ in Eq. (1) was adjusted so that the observed average degree was $<k> = 10.0 \pm 0.1$ in all networks.

the supplementary material in[37] and in[49], the distances between points randomly scattered on the surface of a D-sphere become more homogeneous and approach a constant as $D$ increases, so that the model becomes dependent on the hidden degree only and tends to a zero-clustering limit for very large networks. This is illustrated by the behavior of the average local clustering coefficient of nodes, $C$, and by the mean density of edge triangles, $C_t$, in synthetic networks generated by the $\mathbb{S}^D$ model with realistic parameters, Fig. 1A, B. Specifically, we used power-law distributions for the hidden degrees $P(\kappa) \sim \kappa^{-\gamma}$ and measured $C_t$ as the number of triangles incident on an edge properly normalized. Normalization is performed by dividing the number of triangles going through an edge by the maximum possible number given the degrees at the ends of the edge and then averaged over links that connect nodes in the network with a degree greater than one. The maximum value of both $C$ and $C_t$ is obtained at $\beta = \infty$ (or zero temperature), for which the probability of connection becomes a step function. These maximum values decrease as $D$ increases for all values of the exponent $\gamma$, Fig. 1A, B. As shown, for a typical value of the mean density of edge triangles in real networks, $C_t > 0.5$ (see Table S1), the dimension of the similarity space can be at most $D \approx 7$ when $\gamma = 2.5$, or $D \approx 5$ if $\gamma = 3$. The dependency on dimensionality is also evident for the mean density of (chordless) edge squares, $C_s$, and pentagons, $C_p$, shown in Fig. 1C and 1D. As in the case of triangles, normalization is performed by dividing the number of squares (pentagons) going through an edge by the maximum possible number given the degrees at the ends of the edge discounting participation in triangles (participation in triangles and squares). The obtained quantity is averaged over links that connect nodes in the network with a degree greater than one. As opposed to edge triangles, the densities of edge squares and pentagons are not monotonous but display a maximum for a value of the dimension that increases with network heterogeneity. Notice that clustering coefficients can alternatively be defined relative to individual nodes, instead of edges. While the results are similar, in[50] we found the definition relative to edges to be more stable with respect to degree heterogeneity than the relative to nodes, which is what motivated our choice.

Note that all our measurements in this work are of chordless cycles, to ensure that the densities are not directly dependent on each other. We can therefore exploit this independence by analyzing the "phase space" defined by $(C_t, C_s, C_p)$. In principle, the phase space can

be extended to cycles of higher order. However, in small-world networks, the frequency of chordless cycles above pentagons is extremely low. Figure 2 shows the behavior of pairwise relations between the mean densities of edge triangles, squares, and pentagons in networks generated with the $\mathbb{S}^D/\mathbb{H}^{D+1}$ model for different dimensions $D$ and values of $\gamma$; the graphs reveal the delicate balance between the three densities. For a fixed $D$ and $\gamma$, each curve is obtained by varying the inverse temperature $\beta \in (D, \infty)$. The curves for the different dimensions in Fig. 2A–C, showing the projection on the plane $(C_s, C_t)$, Fig. 2D–F, for the plane $(C_p, C_t)$, and in Fig. 2G–I, for the plane $(C_p, C_s)$, are clearly differentiated, meaning that each dimension presents a characteristic profile.

There are several interesting patterns that can be observed in the phase space $(C_t, C_s, C_p)$. First, all the curves tend to collapse when there is only a small level of clustering, thus becoming dimension independent. This is to be expected because in this case the topological equivalent of the triangle inequality breaks down, so that the network loses its metric character. In addition, all the curves tend to be closer together—and so tend towards dimension independence—as $\gamma \to 2$. This implies that, beyond the fact that a metric space may be needed to explain the observed levels of clustering, its dimensionality is not very important when degrees are strongly heterogeneous and networks are dominated by very big hubs. In turn, this explains why highly heterogeneous real networks are extremely well described by the $\mathbb{S}^1$ model[35,36,41,51].

## Inferring hidden dimensions

These results suggest a method by which we can infer the hidden dimension $D^*$ of a real network. First, we created an ensemble of synthetic surrogates using the $\mathbb{S}^D$ model with different values of the inverse temperature $\beta$ and dimension $D$. To preserve the consistency of the hidden degrees in the synthetic networks and of the observed degrees in the original graph, we computed hidden degrees using an iterative process[35] that forces a match between expected degrees in the model and observed degrees, see subsection "Estimation of the hidden degrees" in the Methods section. This step also ensures that the model reproduces the degree distribution of the real network with high fidelity. After obtaining the hidden degrees, we assigned homogeneous random positions to the $N$ nodes in the $D$-dimensional similarity space. To keep the process computationally efficient, we restricted the ensemble of synthetic surrogates to feasible values of $D$ and $\beta$. The maximum achievable density of edge triangles happens at $\beta = \infty$ and decreases monotonously for increasing values of $D$, as shown above in Fig. 1. Accordingly, we set the maximum dimension to be explored as the maximum dimension able to reproduce the level of edge clustering of the network under study. For each $D$ less than or equal to the maximum value, we decided the set of inverse temperatures, $\beta$, to be explored by sampling homogeneously the phase space of edge clustering in a neighborhood of the observed value, see subsection "Estimation of the range of inverse temperatures in the random ensemble" in Methods. The sample of random surrogates was generated using the connectivity law Eq. (1). The densities of edge triangles, squares, and pentagons were then computed for each surrogate in the ensemble and for the real network. Finally, a data-driven classifier was used to infer $D^*$ from the surrogates that best approximate the real network in terms of edge cycles.

This method can produce as many random network surrogates for a real network as needed to ensure statistical significance. Thus, a simple classifier is suitable for predicting the dimensionality of a network; we used K-nearest neighbors (K-NN). The K-NN classifier identifies the $K$ surrogates closest to the original network in the surrogate $(C_t, C_s, C_p)$ phase space by minimizing Euclidean distance. To maximize the accuracy of our procedure, we also implemented an optimization of parameter $K$ for each original network, see subsection "Classifier selection" in Methods. The inferred dimension of the real network $D^*$ is

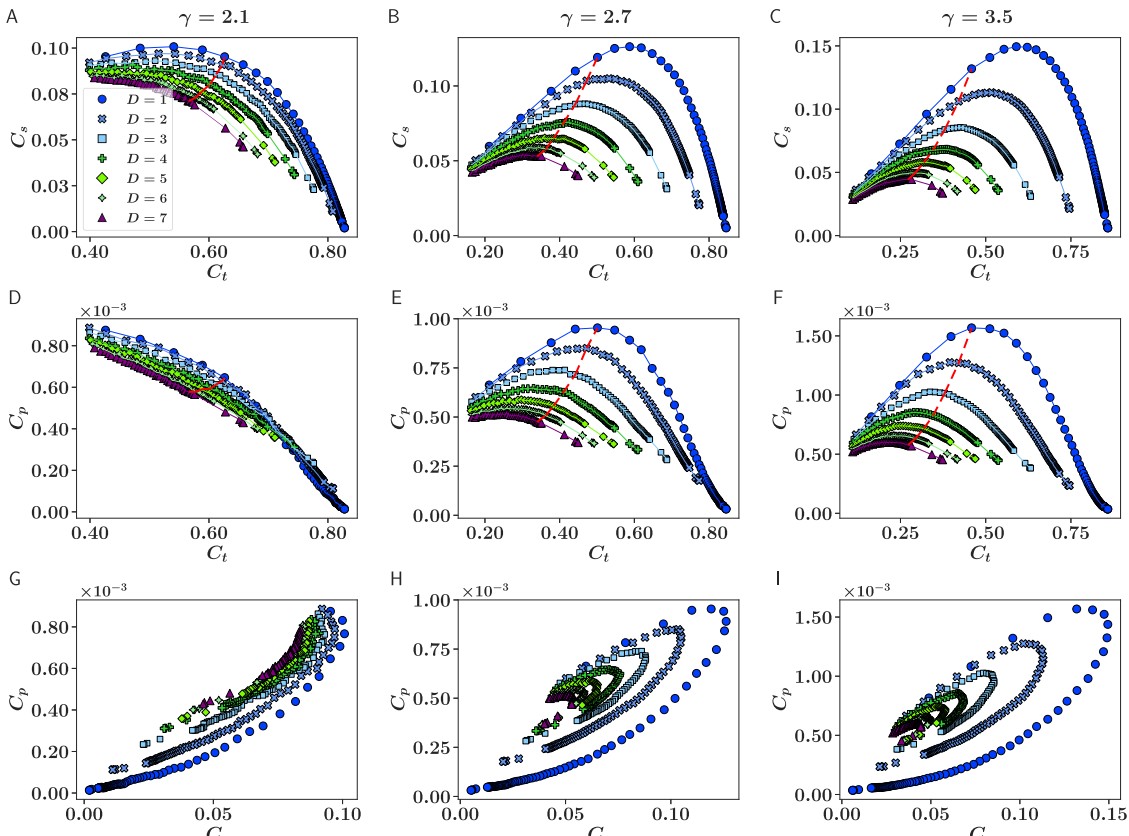

**Fig. 2 | Relation between densities of edge triangles, squares and pentagons for different values of $\gamma$.** Panels (**A–C**) show the projection of the phase space in the subspace $(C_s, C_t)$, panels (**D–F**) show the projection in the subspace $(C_p, C_t)$, and panels (**G–I**) show the projection in the subspace $(C_p, C_s)$. In plots (**A–F**), the dashed red line represents the $\beta = 2D$ limit separating the small-world and large-world phases[34]. In these plots, the area on the left of the dashed red line corresponds to $\beta < 2D$ (small-world phase) and the area on the right corresponds to $\beta > 2D$ (large-world phase if $\gamma > 3$). Each point represents an average over 10 network realizations. Standard errors are smaller than the symbols themselves.

that which maximizes the weighted frequency $f(D) = \sum_{i=1}^{K} \omega_i \delta_{D_i,D}$, where the normalized weights are inversely proportional to the distance $d_i$ between the real network and the $i$-th surrogate in the $(C_t, C_s, C_p)$ space and $\delta_{D_i,D}$ is the Kronecker delta function, so that the most recurrent dimension in neighboring surrogates closest to the original network tends to dominate. We also carried out tests using decision trees and neural networks as classifiers, and obtained similar results, see subsection "Classifier selection" in Methods.

We evaluated the performance of our method by testing it on model networks generated using the $\mathbb{S}^D / \mathbb{H}^{D+1}$ model. Model networks were produced for specific values of $\gamma$, $\beta$, and $D$, and taken as test networks, meaning that an ensemble of random synthetic surrogates was generated for each of them, including the estimation of hidden degrees from observed degrees in the test network. The inferred dimensionality, $D^*$, of the test networks was then used to generate confusion matrices, defined as the probability of predicting $D^*$ in a network generated with dimension $D$. To quantify the goodness of our method, we computed confusion matrices for different types of topologies. Specifically, we tested networks with different degree heterogeneities by varying the exponent $\gamma$, and for each of them we inferred the dimension of networks within two different intervals of $\beta$: one corresponding to the high clustering regime centered around $\beta = 2.5D$, and the other to the low clustering limit in the neighborhood of $\beta = 1.5D$, more details can be found in subsection "Calculation of confusion matrices" in Methods. Confusion matrices for these experiments are shown in Fig. 3. An inference method is considered a good method when the confusion matrices are close to the identity matrix. As shown, the predictions were very good, only generating

mild confusion with contiguous values of $D$ for low values of $\beta$, $\gamma$ and $D$, as expected. This fact is consistent with the tendency of the paired density edge curves $C_t$, $C_S$, $C_p$ to converge for low values of $\beta$ and $\gamma$, as shown in Fig. 2.

## Dimensionality of real networks

We applied our method to infer the dimensionality of real complex networks from different domains. As a case example, Fig. 4 shows mean densities of edge cycles for three real networks: the Internet at the autonomous system level[36], the email network within the Enron company[52], and the human connectome from[53]. The curves for the different dimensions are clearly separated in the email network and in the human connectome, which have very clean estimations of $D^*$ with extremely high accuracies as shown in Table S1 in SI. In both cases, the measured edge densities were well reproduced by the $\mathbb{S}^{D^*}$ model with $D^* = 7$ and $D^* = 3$, respectively. The case of the Internet at the autonomous system level is particularly interesting. Even though the dimension was clearly identified, $D^* = 7$, curves for different dimensions were very close to each other, introducing a certain uncertainty into the inference. This is due to the high heterogeneity of the degree distribution, with a power-law exponent $\gamma \approx 2.1$, which implies that its topology can be reasonable well reproduced even with $D = 1$. Nevertheless, the high value of the inferred dimension suggests that relevant information is hidden in its complex similarity space. This evinces the need to develop embedding algorithms working in arbitrary dimensions that could be used to uncover the hidden attributes of their components.

The cluttering of the curves for the different dimensions in the cycles phase space affects the accuracy of the estimated dimension of a

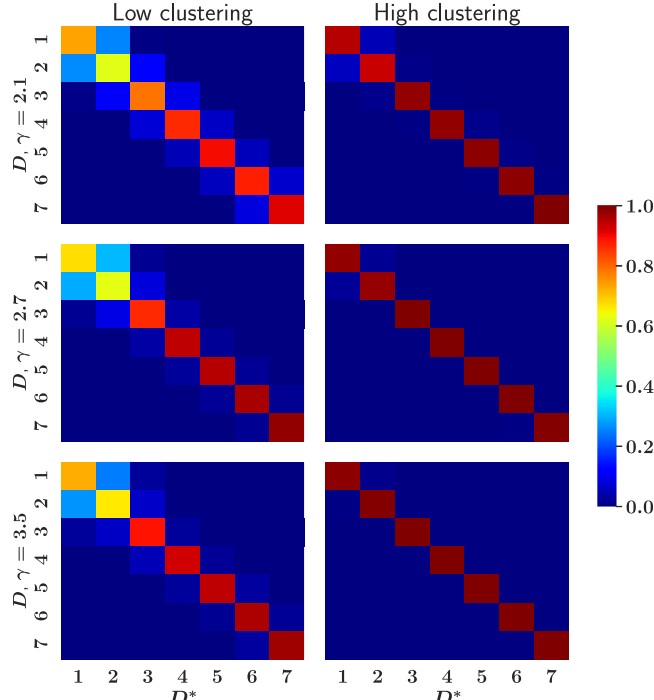

**Fig. 3 | Confusion matrices.** The left column shows results for $\beta = 1.5D$ and different values of $\gamma$. This choice of $\beta$ corresponds to the small-world phase even if $\gamma > 3$. The right column shows the same for $\beta = 2.5D$. In this case, networks with $\gamma > 3$ are large-worlds. The color in each $D - D^*$ box indicates the probability that the predicted dimension is $D^*$ when the dimension used to generate the synthetic network is $D$. Each panel is evaluated with approximately $5 \times 10^3$ networks, more details in sub-section "Calculation of confusion matrices" in Methods.

real network, which is calculated as the proportion of random surrogates of the real network whose dimension is inferred correctly by the corresponding K-NN classifier. Note that accuracy is a measure of the resolution capacity of the dimensionality detection method (error rate associated to the K-NN classifier), which is informative about the error in the best estimate of the embedding dimension because it quantifies how discernible is the dimension of a real network, and the more discernible it is the higher the likelihood that the method selects the best value. Confusion matrices in Fig. 3, defined as the probability of predicting D* in a network generated with dimension D as a function of clustering and heterogeneity in the network, complement the information provided by accuracy. It should be noted that in more than half of the 33 networks analyzed in this work the accuracy reached values of over 90%, and in more than 40% the accuracy is of 99% or absolute.

Figure 5 shows the inferred dimensionality of the 33 real networks as a function of their domain. See also Table S1 where we report the accuracy of the inference. In all the domains, we found most of the networks to have very low to moderate dimensions, in the range $D^* = [1, 4]$. In the higher dimension range, the Internet has an inferred dimension $D^* = 7$, and friendship (both online and offline) and email networks range from $D^* = 6$ up to $D^* = 9$, which could be a reflection of the extremely complex nature of human social interactions. The PGP-trust and EUEmail networks are clear exceptions in the category with $D^* = 1$ and $D^* = 3$, respectively. The former is driven by digital trust between the users of an encryption program, while the later represents email communications between members of a large European research institution. Hence, both these cases depart from the postulates of homophily in social intercourse. Interestingly, the dimensionality of collaboration networks is lower than that of friendship networks, as an indication of more constrained social dynamics in professional environments. In the biological category,

we found one network with higher dimensionality as compared to the others in the same category: the network of genetic interactions in the organism *Drosophila melanogaster*. The higher dimension observed for this network can be understood because it is the projection onto a single layer of a multiplex network that describes different types of genetic interactions. In addition, the dimensionality of brain connectomes remains close to the three dimensions of their anatomical Euclidean embedding, while transport networks are slightly above the two dimensions of their geographical embedding. Finally, the inferred dimensionality for networks of the co-occurrence of terms in language and music is $D^* = 5$ and $D^* = 6$, respectively, while the citation and hyperlink networks are within the intermediate to low dimensionality region.

## Discussion

Dimensionality is a key concept in the project of understanding the geometrical structure of reality. In recent years, the quest to identify dimensionality has reached computer science and network science where, sustained by phenomena of measure concentration, complex data and interactions only populate a small subspace of their original high-dimensional space. The method we present here does not need any a priori spatial embedding and infers the dimensionality of a graph by exploiting the fact that the densities of edge cycles in its topology carry information about its dimensionality. Our results not only prove that complex networks populate a reduced region of a high-dimensional space but also that they are well represented in hyperbolic geometry with ultra low dimensionality.

This claim is valid for complex networks that are simultaneously sparse, small-world, and highly clustered, and possibly—but not necessarily—with heterogeneous degree distributions. Even if a real network is naturally embedded in Euclidean space, this minimal set of complex features ensures that our methodology is applicable, meaning that explicit Euclidean distances in those systems are not the only factor that determines connectivity in the network and other features may have a role as well. At the same time, our formalism is also valid for geometric random graphs, which are very homogeneous and clustered and non small world networks, which happen in our model when $\gamma > 3$ and $\beta > 2D$. This means that graphs in this region of the parameter space are effectively described by the geometry of the $D$-sphere, which becomes the Euclidean space in the thermodynamic limit, rather than by hyperbolic geometry. Finally, links in $D$-dimensional lattices are strongly correlated whereas in our model links are statistical independent. $D$-dimensional lattices are, thus, not suited to being described by our hyperbolic multidimensional model.

We should also mention that some networks present anomalous statistics of higher-order edge cycles that cannot be perfectly matched by the geometric soft configuration model in any dimension. The typical anomalous situation is one in which, for a given value of edge triangles, the value of edge squares or pentagons is higher than the maximum value when dimension $D = 1$. In such cases, our method would predict an inferred dimension $D^* = 1$, but such networks were not included in the datasets we have explored in this work. Also, the $\mathbb{S}^D$ model will tend to give and upper bound to the dimension of real networks with spurious clustering, like in one-mode projections of bipartite networks, but the underestimation is minimized if the network has a heterogeneous degree distribution. Nervertheless, in general the model provides a very good description of the structure of real networks proving that their latent geometry is hyperbolic.

We applied our method to a large number of real networks from different domains and found dimensionalities in the range from one to nine, with some striking regularities. Among these, we found tissue-specific biomolecular networks in the cell to be extremely low dimensional; connectomes in the brain to be close to the three dimensions typical of their anatomical Euclidean embedding; and social and a technological network, such as the Internet, to require

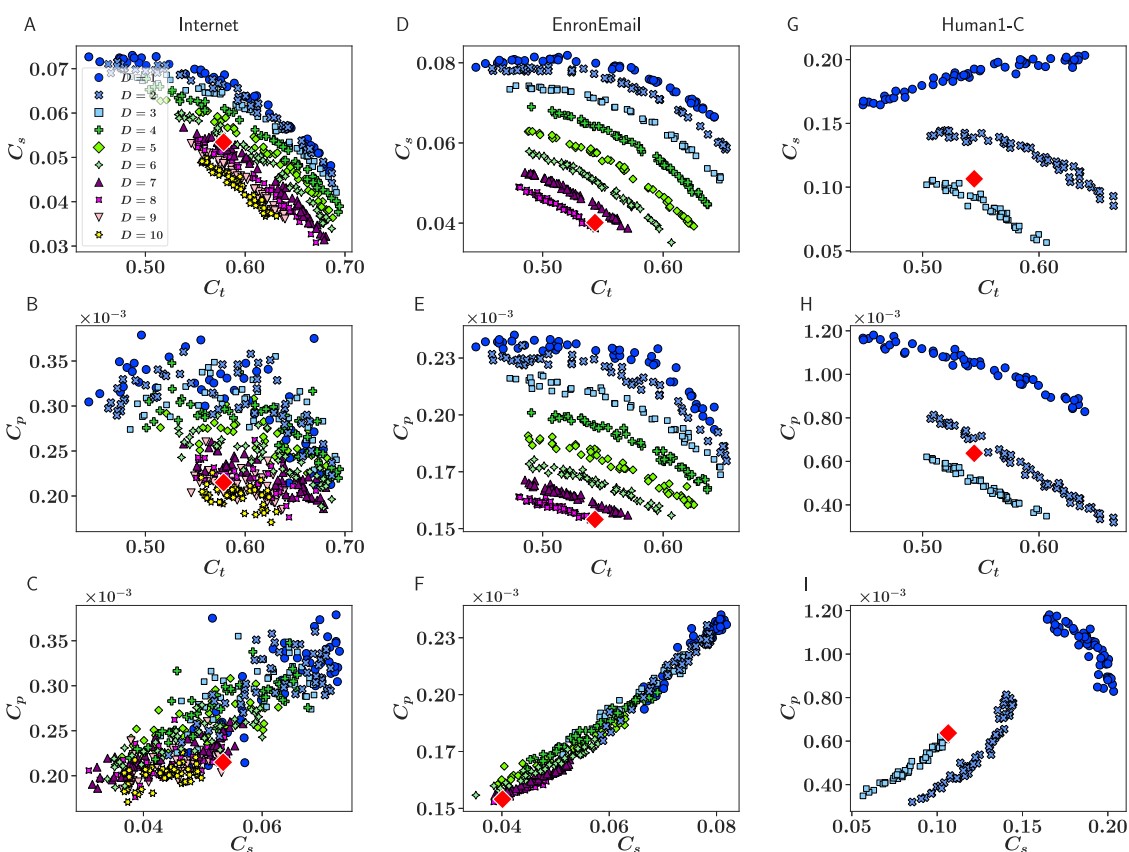

**Fig. 4 | Estimation of the dimension of real networks.** Mean densities of edge triangles, squares, and pentagons for the Internet at the autonomous system level (**A**–**C**), the email network within the Enron company (**D**–**F**), and a human connectome (**G**–**I**).

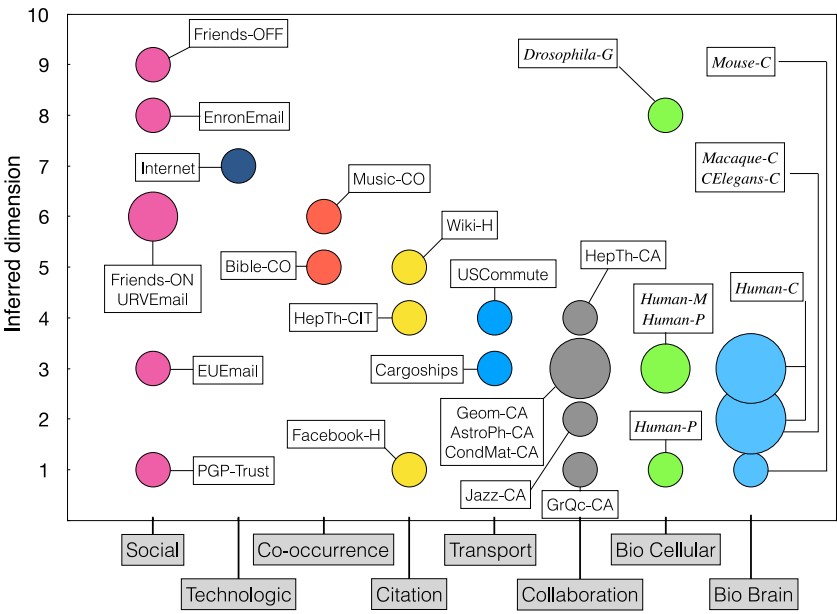

**Fig. 5 | Dimensions of real networks by category.** The size of the symbol is proportional to the number of networks with the same dimension.

more dimensions for a faithful description. This means that despite complex networks having a much lower dimension than their overall size in terms of the number of nodes, more than one similarity dimension is typically needed to map their complex architecture. Social networks based on friendship are at the top of the dimensionality ranking, which reflects homophily in human interactions being determined by a multitude of sociological factors including age, sex,

social class and beliefs or attitudes[54]. Our results are in the range of the logarithmic relationship argued for in ref. 55, between the dimension of an underlying metric space and the number of nodes in online social networks. The case of the Internet is particularly striking. Despite being a technological network, its higher dimensionality is a reflection of the fact that many different factors influence the formation of connections between autonomous systems and, as a consequence, a variety of

relationships may be present, for instance provider–customer, peer-to-peer, exchange-based peering, sibling, or backup relationships, among others[56]. In any case, our estimations are well below estimations based on Euclidean space because hyperbolic space is more appropriate to represent the hierarchical structure of real complex networks and it is more suited to embed real world networks. To make a concrete example, the Internet requires only $D = 7$ dimensions to be embedded in hyperbolic space in our framework, while this number multiplies by six scaling up to $D = 47$ in Euclidean space[57].

Beyond providing a reliable multidimensional hyperbolic model of complex networks that reproduces their structure faithfully with ultra low and customizable dimensionality, our results pave the way towards a meaningful practical method for an ultra efficient dimensional reduction of complex networked system by embedding them in their multidimensional hyperbolic space. Apart from providing more accurate descriptions than two-dimensional maps in the hyperbolic plane, multidimensional hyperbolic embeddings will help to reveal the correlation of factors known to determine connectivity in complex systems—such as geographic and cultural ones in economic and social networks—with the dimensions identified. Another interesting question concerns how the different dimensions in multidimensional hyperbolic maps of real networks line up with multiplexity, where different types of relationships connect pairs of nodes. In addition, our findings can be exploited to create predictive models from relational data with complex structure and help address fundamental issues that hinge on dimensionality. In networks, not only does dimensionality have an impact on the structural characterization of connectivity, but is also crucial for understanding network function, as the dimension governs dynamical processes on networks, such as diffusion and synchronization, as well as their critical behavior.

## Methods

### Estimation of hidden degrees of real networks

Given a real network, our goal is to generate networks with the $\mathbb{S}^D$ model but preserving the degree distribution of the real network as much as possible. In the $\mathbb{S}^D$ model, hidden degrees are given fixed values but nodes' degrees are random variables that depend on the hidden degrees. Therefore, to reproduce the degree distribution, we have to find the sequence of hidden degrees that better reproduces the sequence of observed degrees. To do so, we generalize the method in[35] to arbitrary dimensions. Given a set of parameters $\beta$ and $D$, for each observed degree class $k$ in a real network, we infer the corresponding hidden degree $\kappa$ so that the ensemble average degree of the $\mathbb{S}^D$ model of a node with hidden degree $\kappa$ is equal to its observed degree, that is, $\bar{k}(\kappa) = k$.

After this procedure, the degree distribution of synthetic networks generated by the $\mathbb{S}^D$ model with the inferred sequence of hidden degrees is very similar to the one from the real network. Specifically,

1. Initially set $\kappa_i = k_i \, \forall \, i = 1, N$, where $k_i$ is the observed degree of node $i$ in the real network.
2. Compute the expected degree for each node $i$ according to the $\mathbb{S}^D$ model as

$$\bar{k}(\kappa_i) = \frac{\Gamma\left(\frac{D+1}{2}\right)}{\sqrt{\pi}\Gamma\left(\frac{D}{2}\right)} \sum_{j \neq i} \int_0^\pi \frac{\sin^{D-1}\theta \, d\theta}{1 + \left(\frac{R\theta}{(\mu\kappa_i\kappa_j)^{1/D}}\right)^\beta}, \quad (2)$$

where $R = \left[\frac{N}{2\pi^{\frac{D+1}{2}}}\Gamma\left(\frac{D+1}{2}\right)\right]^{\frac{1}{D}}$ and $\mu = \frac{\beta\Gamma\left(\frac{D}{2}\right)\sin\frac{D\pi}{\beta}}{2\pi^{1+\frac{D}{2}}\langle k \rangle}$.

3. Correct hidden degrees: Let $\epsilon_{max} = \max\{|\bar{k}(\kappa_i) - k_i|\}$ be the maximal deviation between actual degrees and expected degrees.
   - If $\epsilon_{max} > \epsilon$, the set of hidden degrees needs to be corrected. Then, for every class of degree $k_i$, we set $|\kappa_i + [k_i - \bar{k}(\kappa_i)]u| \to \kappa_i$, where $u$ is a random variable drawn from $U(0, 1)$. The random variable $u$ prevents the process from getting trapped in a local

minimum. Next, go to step 2 to compute the expected degrees corresponding to the new set of hidden degrees.
   - Otherwise, if $\epsilon_{max} \leq \epsilon$, hidden degrees have been correctly inferred for the current global parameters.

Following this algorithm, we can generate surrogates of a given network $G$ with different $D$ and $\beta$ values without modifying the degree distribution. The tolerance value of $\epsilon$ used in this work is $\epsilon = 1$.

### Estimation of the range of inverse temperatures in the random ensemble

Inverse temperature $\beta$ can take any value within the range $(D, \infty)$. However, the relation between edge clustering $C_t^D(\beta)$ and $\beta$ in the $\mathbb{S}^D$ model is non-linear, approaching zero at $\beta \to D^+$ (in the thermodynamic limit) and converging to a constant value when $\beta \to \infty$. Our aim is to sample homogeneously the phase space of edge clustering in a neighborhood of the observed value. That is, if $C_t^*$ is the observed edge clustering of a real network, we have to generate surrogate networks homogeneously within the interval $(C_t^* - \Delta C_t, C_t^* + \Delta C_t)$.

We first focus on the one dimensional case $D = 1$. In this case, we first generate a small sample of 20 networks with the $\mathbb{S}^1$ model for different values of $\beta$ drawn from an uniform distribution $U(1, 15)$ and perform a non-linear fitting to the obtained values as a function of $\beta$ with the function

$$C_t^1(\beta) = C_{t,max}^1(1 - e^{-a(\beta - \beta_0)}), \quad (3)$$

where $C_{t,max}^1$, $a$, and $\beta_0$ are fitting parameters. Once the fitting parameters are know, we can invert Eq. (3) to obtain $\beta$ as a function of $C_t^1$ as

$$\beta = \beta_0 - \frac{1}{a}\ln\left(1 - \frac{C_t^1}{C_{t,max}^1}\right). \quad (4)$$

Then, to generate the final set of networks, we sample values uniformly in the interval $\xi \in (C_t^* - \Delta C_t, C_t^* + \Delta C_t)$ and use Eq. (4) with $C_t^1 = \xi$ to get the corresponding values of $\beta$. In this work we use $\Delta C_t = 0.1$.

For higher dimensions, we make use of the approximate empirical scaling relation

$$\frac{C_t^D(\beta/D)}{C_{t,max}^D} \approx \frac{C_t^1(\beta)}{C_{t,max}^1}, \quad (5)$$

which implies that the functional dependence of edge clustering on $\beta$ is approximately the same for every dimension up to scaling factors, see Supplementary Fig. S1. This allows us to extrapolate the range of values of $\beta$ to be explored. Specifically, as in the one dimensional case, we sample values uniformly in the interval $\xi \in (C_t^* - \Delta C_t, C_t^* + \Delta C_t)$. Then, for each sampled value $\xi$ we associate a value of $\beta$ as

$$\beta = D\left[\beta_0 - \frac{1}{a}\ln\left(1 - \frac{\xi}{C_{t,max}^1}\right)\right], \quad (6)$$

where parameters $a$ and $\beta_0$ and $C_{t,max}^1$ are the ones obtained for the case $D = 1$. Notice, however, that in some cases it is not possible to sample the same interval of edge clustering in all explored dimension. This can be due to the imperfect scaling relation Eq. (5), to the fact that $C_t^* + \Delta C_t$ is higher than the value that can be reproduced in a given dimension, or that $C_t^* - \Delta C_t$ gives values of $\beta$ below the minimum allowed value, that we set to $D + 0.25$. Using this procedure, we generate a set of 50 networks per dimension.

### Classifier selection

In order to learn the relation between the proportion of triangles, squares and pentagons in a real network and its dimensionality we

have evaluated different supervised machine learning (ML) techniques. Our examples are copies generated using the adjusted $\mathbb{S}^D$ model and we use cycle proportions as predictors and the dimension as the target property.

Let us consider the set of copies $\mathbf{x} = (\phi_1, ..., \phi_N)$, each copy $\phi$ characterized by a vector $\phi = [C_t(\phi), C_s(\phi), C_p(\phi)]$ of 3 features and dimension $D(\phi)$. Being $D^*(\phi)$ the target feature to be estimated, the problem consists of finding the function $f$ such that $f(\phi) = D(\phi) \pm \Delta D$ that minimizes $\Delta D = |D^*(\phi)\text{-}D(\phi)|$ for every $\phi$ in $\mathbf{x}$. Since the dimension of each copy is known in the training step, we use supervised ML techniques to estimate function $f$.

In our problem, the target estimation $D$ is a discrete value. Although, there exist many different ML algorithms to predict this kind of values, the no-free-lunch theorem[58] states that no ML algorithm is the best for every problem. In fact, the performance of ML algorithms strongly depends on the classification problem, and the number and distribution of the input instances. Therefore, it is unknown a priori the techniques showing a good performance, and finding them usually requires a trial and error process.

When selecting the machine learning method that allows to infer the dimensionality associated to a network following the process described in this paper, we have carried out tests with neural networks[59], decision trees[60] and K-nearest neighbors (K-NN)[61] methods. In the case of neural networks, we used a feed-forward architecture (FNN) with a 64-neurons hidden layer and *Adam*[62] as optimizer (hyperbolic tangent as the activation function). In the case of K-NN, the target is predicted by local interpolation of the targets associated to the nearest neighbors in the training set. The decision tree uses a tree-like model of queries that allow to classify an instance. Supplementary Fig. S2 illustrates the decision regions learned by each of these three methods when generalizing the dimensionality of a set of networks. For the sake of clarity, only the proportion of triangles and squares have been taken into account. We have obtained very good results with K-NN and neural networks.

At the light of these results, we have selected the K-NN method due to its simplicity and geometric interpretation. The implementation of machine learning methods (including K-NN) used in our experiments are those found in[63]. Finally, to maximize the accuracy of our method, we select the value of the parameter $K$ as the one maximizing the proportion of synthetic networks correctly classified with respect to their dimension. This value is then used to infer the dimension of the original network using the K-NN method.

## Calculation of confusion matrices
Tables 1 and 2.

**Table 1 | Intervals of $C_t$ for $D = 1$ used to evaluate confusion matrices for different regions of clustering and levels of heterogeneity**

|                 | $\gamma = 2.1$ | $\gamma = 2.7$ | $\gamma = 3.5$ |
|-----------------|----------------|----------------|----------------|
| Low clustering  | (0.28, 0.55)   | (0.18, 0.39)   | (0.14, 0.35)   |
| High clustering | (0.43, 0.77)   | (0.27, 0.69)   | (0.23, 0.65)   |

**Table 2 | Number of surrogates used to evaluate confusion matrices for different regions of clustering and levels of heterogeneity**

|                 | $\gamma = 2.1$ | $\gamma = 2.7$ | $\gamma = 3.5$ |
|-----------------|----------------|----------------|----------------|
| Low clustering  | 5215           | 5215           | 5215           |
| High clustering | 5215           | 3874           | 3725           |

## Reporting summary
Further information on research design is available in the Nature Research Reporting Summary linked to this article.

## Data availability
All the data or their source reference are available in the manuscript and the supplementary materials, or they will be provided upon request.

## Code availability
The codes for the computation of cycles in real networks and for the implementation of $\mathbb{S}^D$ surrogates can be accessed in the GitHub public repository online at https://github.com/networkgeometry/detecting-dimensionality, and can be cited as ref. 64.

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

## Acknowledgements

We thank Robert Jankowski for a careful reading of the manuscript. We acknowledge support from: Agencia estatal de investigación project number PID2019-106290GB-C22/AEI/10.13039/501100011033; project Mapping Big Data Systems: embedding large complex networks in low-dimensional hidden metric spaces, Ayudas Fundación BBVA a Equipos de Investigación Científica 2017; M.A.S. and M.B. acknowledge support from Generalitat de Catalunya grant number 2017SGR1064. M.B. acknowledges support from the ICREA Academia award, funded by the Generalitat de Catalunya.

## Author contributions

M.A.S. and M.B. designed research. P.A. performed computations. All authors participated in the implementation of the research, analysis of results, and in the writing of the paper.

## Competing interests

The authors declare no competing interests.
