## [Peer Review File · Nature Communications]

REVIEWER COMMENTS

Reviewer #1 (Remarks to the Author):

The manuscript by Almagro and collaborators deals with a new methodology for estimating of the embedding dimension of real-world networks. The procedure relies on fitting real networks against the generalization to arbitrary D dimensions of a model introduced in earlier by the senior authors of the manuscript. The fitting procedure aims at optimizing the match between observed topological properties of a real graph and those of the best model. Properties considered in the fit regard densities of cycles of length 3, 4 and 5. The main finding of the paper is that real-world networks can be embedded in extremely low-dimensional spaces. The least parsimonious embedding is the one obtained for a snapshot of the Internet at the autonomous system level, which can be optimally embedded in $D=8$ dimensions. All other networks considered in the study are embedded in lower-dimensional spaces, typically in the range $D=1$ to $D=3$.

The topic of the paper is very important for current research in network science. The methodology is original and sound. Results of the paper are quite interesting. Points of strength of the manuscript are:

1. The methodology is very clever as it does not require to actually perform the embedding of a network in order to estimate its optimal embedding dimension. I particularly liked the idea of using short cycles as main ingredients in the recipe proposed by the authors.
2. The findings from the analysis of real-world networks are particularly impressive especially in view of results reported in recent literature. It seems that hyperbolic space is very well suited to embed (some) real world networks compared to Euclidean space. To make a concrete example, the AS-level Internet requires $D=8$ dimensions to be embedded in hyperbolic space according to the present analysis, but $D=47$ in Euclidean space [see for example Phys. Rev. E 103, 012305 (2021)].

In summary, I believe that the manuscript deserves space in the journal. However, before recommending the paper for publication, I would like that see the following issues addressed:

1. What does it happen if the approach is applied to networks are naturally embedded in the Euclidean space, like a D -dimensional lattices or transportation (e.g., road, train) networks? Are these cases where we expect a methodology which intrinsically relies on hyperbolic geometry to fall short? A missing part of the current version of the manuscript is a discussion about the limitations of proposed methodology. Honestly, I would not consider negative cases as bad; however, they may allows us to identify types of graphs where the proposed approach would be not ideal.
2. No discussion about the computational complexity of the method is presented. I see that the authors considered networks with sizes up to $N=30,000$. Is the approach applicable to networks with $N=10^6$ nodes? If not, why not? Also, how long does it take to properly estimate the dimension of a network with $N=30,000$ nodes? If the method does not require to actually perform the embedding, and topological measurements regard only local cycles, I do not see why the method can not easily scale up to large networks.

3. The actual fitting procedure is quite involved, and I didn't follow all the details. However, I noted that some of the visualizations reported in the SM look qualitatively different from those reported in the main text. I refer to all cases in Figures S1-S6 where the red diamond, which refers to the actual real network at hand, is well outside the cloud of points that represent instances of the S^D model. Examples are: condmat-CA, Drosophila-G, Geom-CA, GrQc-CA, Human2-P, Wiki-H, etc. In particular, this observation seems to be systematically valid for all the collaboration networks considered in the study. What does it mean? Can we say that the S^D model well capture the topological properties of the networks in these cases too? Also, how should we interpret the plots for Human3-C, Jazz-CA and Music-CO, where model instances correspond to vertical stripes of points?

4. Would it make sense to define a sort of p-value to estimate the goodness of the fit? Also, does the procedure allow to associate an error to the best estimate of the embedding dimension? This would be a valuable addition to the study, being the methodology based on statistical sampling (thus automatically leading to the notion of statistical significance) and aimed at dealing with empirical data (thus requiring an estimate of an error).

5. In reference to points 3 and 4, the authors state in the conclusions of the paper that "...some networks present anomalous statistics of higher-order edge cycles that cannot be perfectly matched by the S^D model in any dimension, although in general the model provides a very good description of the structure of real networks and proves that their latent geometry is hyperbolic." What networks were excluded from the analysis? How was the quality of the embedding quantified? Also, how do the authors justify the statement that "complex networks ... are better represented in hyperbolic geometry with ultra low dimensionality." What is the term of comparison here?

6. It would be great to consider an alternative way to validate the results of the paper. For instance, the current methodology determines the value of D^* of a network by fitting it against the S^D model. Does this translate into optimal performance in network tasks that rely on the S^D embedding? Specifically, does the D^* -dimensional embedding of the network provide superior performance compared to a D -dimensional embedding, with $D \neq D^*$? The authors could consider performance in greedy navigation, a quite popular application of hyperbolic network embedding.

Minor concerns:

- 1. Questions marks appear in place of reference numbers in the Methods section.**
- 2. I would increase the font size in the various figures to improve readability.**

Reviewer #2 (Remarks to the Author):

This paper tries to find best parameters of a generative model constructed from hyperbolic geometry principles to match a given network using averaged combinatorial (local) measures, then propose one of these parameters, the dimension, to be a meaningful notion of dimension of the original network.

I have several questions about this approach and its usefulness in the context of network science.

- The obtained notion of dimension is an integer number for a whole graph, thus a very simplistic measure. Many widely different graphs can have this same D , so what do we learn from this measure specific about the graph? Most interesting measures on graphs are based on nodes, and global measures may be used to compare/classify graphs, but cannot be rigid -- they need to capture what is different between given graphs.**
- The fit to obtain D is an average of local quantities, but complex networks may be very heterogeneous, thus this average value may not be representative at all of the graph structure (take a graph with two different regions with different structures, which will make distribution of some of the combinatorial measures bimodal, the mean value). So I believe that applying dimension reduction methods using this dimension as is proposed would not work on such graphs. Thus the use of this dimension for dimensionality reduction methods may be fairly limited to simple graphs.**
- It is claimed that there is no need for spatial embedding, but the dimension of the hyperbolic space can be seen as an embedding, so this method can be seen as an 'undirect hyperbolic spatial embedding', highly dependent on the choices of the combinatorial measures used to find the best hyperbolic surrogate. It may be that these measures are the most relevant for this task, but it makes the notion of dimension depend on several non-trivial modeling choices, and therefore cannot be shown to be a 'fundamental measure', as one expect from the natural notion of dimension.**
- I am not familiar enough with generating these hyperbolic surrogates, but I don't believe it is such an efficient strategy to generate many models, and find the closest match to estimate a single integer number, as is claimed.**
- There is no study of comparing this dimension measure with previous ones, or other global graphs quantities (radius, mass, etc...) to show that D provides a non-trivial measure that cannot be obtained via other much simpler computations.**
- There is no analysis of what the dimension of simple geometric graphs would be, such as a grid/meshes. Would we get $D=2$ for a simple grid graph, and $D=1$ for a line graph? I would expect a meaningful measure of dimension would satisfy this basic concept.**

Overall, I don't think this paper is suited for Nature Communication, but rather for a more specialized network science journal, provided some of the points above are addressed.

Resubmission of manuscript NCOMMS-21-42385

Detecting the ultra low dimensionality of real networks

Pedro Almagro, Marián Boguñá, M. Ángeles Serrano

Reviewer #1

Comment 1.0.: The manuscript by Almagro and collaborators deals with a new methodology for estimating of the embedding dimension of real-world networks. The procedure relies on fitting real networks against the generalization to arbitrary D dimensions of a model introduced in earlier by the senior authors of the manuscript. The fitting procedure aims at optimizing the match between observed topological properties of a real graph and those of the best model. Properties considered in the fit regard densities of cycles of length 3, 4 and 5. The main finding of the paper is that real-world networks can be embedded in extremely low-dimensional spaces. The least parsimonious embedding is the one obtained for a snapshot of the Internet at the autonomous system level, which can be optimally embedded in $D=8$ dimensions. All other networks considered in the study are embedded in lower-dimensional spaces, typically in the range $D=1$ to $D=3$.

The topic of the paper is very important for current research in network science. The methodology is original and sound. Results of the paper are quite interesting. Points of strength of the manuscript are:

1. The methodology is very clever as it does not require to actually perform the embedding of a network in order to estimate its optimal embedding dimension. I particularly liked the idea of using short cycles as main ingredients in the recipe proposed by the authors.
2. The findings from the analysis of real-world networks are particularly impressive especially in view of results reported in recent literature. It seems that hyperbolic space is very well suited to embed (some) real world networks compared to Euclidean space. To make a concrete example, the AS-level Internet requires $D=8$ dimensions to be embedded in hyperbolic space according to the present analysis, but $D=47$ in Euclidean space [see for example Phys. Rev. E 103, 012305 (2021)].

In summary, I believe that the manuscript deserves space in the journal.

Reply: We thank the reviewer for their time and effort in reviewing our work and for their very positive opinion. We are specially pleased that they understood perfectly the strengths of our contribution and we really appreciate their opinion that our manuscript deserves space in Nature Communications.

We are also very grateful for all the constructive comments and suggestions that have certainly helped us to improve the quality and presentation of our manuscript, and also for providing us with the reference above that is relevant to our work. In this resubmission, we have carefully addressed all of their comments and hope that the manuscript is suitable for publication.

Action taken: We have added a citation to Phys. Rev. E 103, 012305 (2021).

Comment 1.1.: However, before recommending the paper for publication, I would like that see the following issues addressed:

1. What does it happen if the approach is applied to networks are naturally embedded in the Euclidean

space, like a D -dimensional lattices or transportation (e.g., road, train) networks? Are these cases where we expect a methodology which intrinsically relies on hyperbolic geometry to fall short? A missing part of the current version of the manuscript is a discussion about the limitations of proposed methodology. Honestly, I would not consider negative cases as bad; however, they may allow us to identify types of graphs where the proposed approach would be not ideal.

Reply: Hyperbolic geometry is a meaningful framework for complex networks that are simultaneously sparse, small-world, and highly clustered, and possibly—but not necessarily—with heterogeneous degree distributions. If a real network naturally embedded in Euclidean space presents this minimal set of complex features (even if the power of our geometric models reach far beyond those properties to explain also community structure, self-similarity, navigability, the triangle inequality violation spectrum in weighted networks...), which is the most probable situation since they are ubiquitous and characterize the complexity of interactions in real complex systems, then our methodology is definitely applicable. This means that explicit distance in complex networks naturally embedded in Euclidean space may not be the only factor that determines connectivity in those network and other features may have a role as well.

One prototypical example is that of brain anatomy, sustained by neural networks whose architecture has been shaped in Euclidean space by physical constraints and communication needs throughout evolution. However, despite the brain being naturally embedded in 3-D Euclidean space and even if Euclidean space is typically assumed as its natural geometry, distances in hyperbolic space offer a more accurate interpretation of the connectivity structure of connectomes, which suggests a new perspective for the mapping of the brain's neuroanatomical regions. We proved and intensively discussed this in previous works, see Refs. [38] and [40] (in the numbering of the new version of the manuscript). In the present manuscript, among the considered real networks we analyzed 9 brain connectomes of 4 species including humans, and in all cases the dimensionality in hyperbolic space displays values between $D=1$ and $D=3$, remaining close to the three dimensions of their anatomical Euclidean embedding. Another spatial network that we analyzed in our work is USCommute, a transportation network that reflects the daily commuter traffic between US counties. In this case, the optimal dimensionality in the hyperbolic description is $D=4$, meaning that other factors apart from geographical distances are expected to have a role in explaining its connectivity.

At the same time, our formalism is also valid for geometric random graphs, which are very homogeneous and clustered and non small world networks, which happens in our model when $\gamma > 3$ and $\beta > 2D$. This means that graphs in this region of the parameter space are effectively described by the geometry of the D -sphere, which becomes the Euclidean space in the thermodynamic limit, rather than by hyperbolic geometry.

Finally, D -dimensional lattices are clearly out of the category of complex networks and of geometric random graphs. The position of their nodes in the space is highly ordered and their links are strongly correlated whereas in our model links are statistical independent. D -dimensional lattices are, thus, not suited to being described by our hyperbolic multidimensional model.

Action taken: We have included in section “Discussion” a paragraph to clarify the limitations of the proposed methodology where we also discuss the situation of networks with explicit geometry.

Comment 1.2.: 2. No discussion about the computational complexity of the method is presented. I see that the authors considered networks with sizes up to $N=30,000$. Is the approach applicable to networks with $N = 10^6$ nodes? If not, why not? Also, how long does it take to properly estimate the dimension of a network with $N=30,000$ nodes? If the method does not require to actually perform the embedding, and topological measurements regard only local cycles, I do not see why the method can

not easily scale up to large networks.

Reply: The computational complexity in our work is actually quite important because we analyze a large number of different real networks. However, for a given particular network, we could reach networks up to 10^6 nodes, although it could take a significant amount of computational time. The limiting step in the process is the generation of surrogates. To be more precise, for a given network, we have to generate surrogates with the same degree sequence for a range of values of β and for different dimensions. The computational complexity of each such surrogate is of the order N^2 , with N the size of the network. We first need to determine the maximum dimension to be tested. This step can be performed easily because the maximum average clustering coefficient of our model decreases with the network dimension. Thus, it is enough to generate surrogates at $\beta = \infty$ increasing the dimension until the generated clustering coefficient in synthetic surrogates is smaller than the observed one. Then, we generate surrogates for each dimension below the maximum in an appropriate range of values of β . A reasonable number of surrogates per dimension is between 10 and 20. If, for instance, the maximum dimension is 5, the number of surrogates to be generated would be between 50 and 100. The physical time to generate a network of 10^6 nodes and standard average degree is around 15 minutes in a standard computer. Thus, in our example we would need between 12 and 24 hours of computation physical time. Of course, this time could be reduced significantly by proper parallelization of the workflow and an optimal estimation of the range of β to be explored. However, the present work is a proof of concept that our method can be applied to real networks rather than to develop the fastest algorithm to detect the dimension.

Comment 1.3.: 3. The actual fitting procedure is quite involved, and I didn't follow all the details. However, I noted that some of the visualizations reported in the SM look qualitatively different from those reported in the main text. I refer to all cases in Figures S1-S6 where the red diamond, which refers to the actual real network at hand, is well outside the cloud of points that represent instances of the \mathbb{S}^D model. Examples are: condmat-CA, Drosophila-G, Geom-CA, GrQc-CA, Human2-P, Wiki-H, etc. In particular, this observation seems to be systematically valid for all the collaboration networks considered in the study. What does it mean? Can we say that the \mathbb{S}^D model well capture the topological properties of the networks in these cases too? Also, how should we interpret the plots for Human3-C, Jazz-CA and Music-CO, where model instances correspond to vertical stripes of points?

Reply: We start answering the question of the model instances corresponding to vertical stripes of points in graphs Ct-Cp of the phase space, which is the most important remark in the referee's comment. We thank the referee for making us realize that we were in fact measuring a subclass of chordless pentagons (those with nodes that do not participate in squares) instead of the total amount. We have recalculated the densities of pentagons to take into account all possible chordless cycles of size 5 and not only the subclass considered in the previous version of the manuscript, which led to results that were correct (consistent with the restriction on pentagons) but difficult to interpret and non-informative as the referee pointed out. This change has led to new versions of all the figures in the manuscript and supporting information containing densities of pentagons. In the new figures, we do not find vertical stripes in the graphs Ct-Cp for any network, which now show patterns resembling those in the Ct-Cs graphs. We have repeated the estimation of the dimensionality for all real networks and, qualitatively, the results have not changed as shown in the new version of Fig. 5 in the manuscript. Quantitatively, the dimensionalities of some real networks have changed ± 1 around the reported values in the first version of our manuscript and, in general, the accuracies have improved.

Regarding the first part of the comment, the red diamond, which refers to the actual real network at hand, can be outside the cloud of points that represent instances of the \mathbb{S}^D model for different reasons. In the networks reported in our work, the reason is basically related with the quality or biases of the network reconstructions. First, notice that collaboration networks are originally bipartite networks of

scientists and papers and that the process of obtaining one-mode projections of scientists typically generates some amount of spurious clustering. As a result, these networks present an anomalous large amount of clustering and, since we are considering chordless cycles, this also implies a decreased low number of other higher order cycles, such as squares and pentagons. The maximum attainable clustering in the \mathbb{S}^D model decreases with dimensionality and, in the phase space, we only included curves for dimensions that were able to reach the level of clustering of the real network under study. In any case, the \mathbb{S}^D model will tend to give an upper bound to the dimension of these real networks but spurious clustering, and so this underestimation, is minimized if the network has a heterogeneous degree distribution.

Networks with very low values of clustering can also fall outside the cloud of points that represent instances of the \mathbb{S}^D model, such as Human2-P. To keep the process computationally efficient, we restrict the ensemble of synthetic surrogates of a real network to feasible values of β and D and perform a systematic estimation of the range of inverse temperatures and maximum dimensionality in the random ensemble as explained in Methods. Notice, however, that in some cases it is not possible to sample the same interval of edge clustering in all explored dimension or that clustering gives values of β below the minimum allowed value, that we set to $D + 0.25$. In any case, a low value of clustering indicates that the network itself has weak metric properties and then the notion of dimensionality becomes diluted. Finally, we would not say that Wiki-H fall outside the \mathbb{S}^D phase space. As for Drosophila-G, the empirical densities of triangles and squares fall within the cloud of points and only deviates for the density of pentagons.

Action taken: We have recalculated the densities of pentagones to take into account all possible chordless cycles of size 5. This change has led to new versions of all the figures in the manuscript and supporting information containing densities of pentagons. We have reversed the axes in the representation of triangles versus pentagones for consistency with the representation of triangles versus squares. We have repeated the estimation of the dimensionality for all real networks. Accordingly, we have adjusted the text in the manuscript.

We also added a clarification of the limitations of the proposed methodology in the section “Discussion” concerning the impact of data quality in the amount of cycles in network reconstructions and possible biases introduced by spurious clustering in one-mode projections of originally bipartite networks.

Comment 1.4.: 4. Would it make sense to define a sort of p-value to estimate the goodness of the fit? Also, does the procedure allow to associate an error to the best estimate of the embedding dimension? This would be a valuable addition to the study, being the methodology based on statistical sampling (thus automatically leading to the notion of statistical significance) and aimed at dealing with empirical data (thus requiring an estimate of an error).

Reply: In the first version of our manuscript, we were already providing the accuracy of the K-NN classifier (the complementary of its error rate) as a measure of the resolution capacity of the method, we are sorry that this information was not clear enough. This quantity was calculated as the proportion of random surrogates of the real network—synthetic networks produced with the \mathbb{S}^D model with exactly the same degree distribution as the real network—whose dimension is inferred correctly by the corresponding K-NN classifier. Typically, the value is high when the curves for pairwise relations in the cycles phase space are well separated. In the Table S1 contained in the Supplementary Information file we report the accuracy value for each real network analyzed in our work. It should be noted that in more than half of the 33 real networks, the accuracy reached values of over 90%, and in more than 40% of the networks the value of accuracy was above 99%.

Accuracy is informative about the error in the best estimate of the embedding dimension because it

quantifies of how discernible is the dimension of a real network, and the more discernible it is the higher the likelihood that the method selects the best value. Confusion matrices in Fig. 3, defined as the probability of predicting D^* in a network generated with dimension D as a function of clustering and heterogeneity in the network, complement the information provided by accuracy. Figure 3 shows that the dimension of a complex network is clearly discernible in general except for a mild level of confusion in contiguous values of D when clustering is low and the exponent of the degree distribution is close to -2 . For instance, in the case of the Internet the accuracy value is 66%, consistent with the fact that the exponent of its degree distribution is close to -2 , meaning that more than half of the times the method infers the correct dimension and in the cases where it does not so the inferred dimension is typically one unit off, 7 or 9. However, in such cases the cluttering of curves for pairwise relations in the cycles phase space implies that the error made by selecting the wrong dimension becomes less important because the topological features of networks with adjacent dimensions become very similar.

Action taken: In the main text, we have clarified the definition of accuracy as a measure of the resolution capacity of the dimensionality detection method (error rate associated to the K-NN classifier), and we have included an explanation of how, when combined with the information provided by the confusion matrices in Fig. 3, it is informative about the error measure to the best estimate of the embedding dimension.

Comment 1.5.: 5. In reference to points 3 and 4, the authors state in the conclusions of the paper that "...some networks present anomalous statistics of higher-order edge cycles that cannot be perfectly matched by the \mathbb{S}^D model in any dimension, although in general the model provides a very good description of the structure of real networks and proves that their latent geometry is hyperbolic." What networks were excluded from the analysis? How was the quality of the embedding quantified? Also, how do the authors justify the statement that "complex networks ... are better represented in hyperbolic geometry with ultra low dimensionality." What is the term of comparison here?

Reply and action taken: Basically, the networks excluded from the analysis fall into two categories: networks of genetic interactions and airport networks. As was explained in the manuscript, the typical anomalous situation is one in which, for a given value of edge triangles, the value of edge squares is higher than the maximum value when dimension $D = 1$. In such cases, our method would predict an inferred dimension $D = 1$. In other words, such networks present an anomalous high value of squares.

We agree that the sentence, "in general the model provides a very good description of the structure of real networks and proves that their latent geometry is hyperbolic", was misplaced and we have removed it. What we mean with this sentence is that the \mathbb{S}^D model provides a very good description of the structure of the majority of real networks that we analyzed in our work. This is not measured by any embedding but it is implied from the fact that the phase space of cycles for each network shows that, given their degree distribution, the model can reproduce well not only their sparsity, small world property, and level of clustering—as already proven in previous publications by some of us—but also more sophisticated features such as the statistics of higher order cycles. To avoid confusion, we have changed the word "better" by "well".

Comment 1.6.: 6. It would be great to consider an alternative way to validate the results of the paper. For instance, the current methodology determines the value of D^* of a network by fitting it against the \mathbb{S}^D model. Does this translate into optimal performance in network tasks that rely on the \mathbb{S}^D embedding? Specifically, does the D^* -dimensional embedding of the network provide superior performance compared to a D -dimensional embedding, with $D \neq D^*$? The authors could consider performance in greedy navigation, a quite popular application of hyperbolic network embedding.

Reply: We cannot validate our results against any task that requires the embedding of the real networks

into the \mathbb{S}^D hyperbolic geometry, since one of the strong points of our methodology to detect the dimensionality of a real network is that it does not require an embedding. We based our methodology in preserving the degree distribution and selecting the dimension of the model that best reproduces the topological statistics of the network in terms of cycles. So, multidimensional embedding methods are beyond the scope of the present manuscript. As a consequence, we cannot test greedy navigation or any other task that requires a map of the real networks in the underlying latent metric space. Let us just add here that we are already working on such a methodology to embed real networks into the \mathbb{S}^D multidimensional hyperbolic space and it is far from trivial to find the efficient and computationally feasible algorithm.

Comment 1.7.: Minor concerns:

1. Questions marks appear in place of reference numbers in the Methods section.
2. I would increase the font size in the various figures to improve readability.

Reply and action taken: Thank you for pointing this out. We have corrected these issues.

Reviewer #2

Comment 2.0: This paper tries to find best parameters of a generative model constructed from hyperbolic geometry principles to match a given network using averaged combinatorial (local) measures, then propose one of these parameters, the dimension, to be a meaningful notion of dimension of the original network.

Reply: We thank the reviewer for their time and effort in reading our paper and for their report, which has helped us to improve substantially the presentation of our results.

Comment 2.1: I have several questions about this approach and its usefulness in the context of network science.

- The obtained notion of dimension is an integer number for a whole graph, thus a very simplistic measure. Many widely different graphs can have this same D , so what do we learn from this measure specific about the graph? Most interesting measures on graphs are based on nodes, and global measures may be used to compare/classify graphs, but cannot be rigid – they need to capture what is different between given graphs.

Reply: The goal of our work is not to characterize a real network by a simplistic measure. Far from characterizing a real complex network with a single integer number, our methodology characterizes the network using the geometric soft configuration model, or \mathbb{S}^D model, which is a geometric random graph model that preserves the original degree distribution and such that the dimensionality can be adjusted to the optimal value to reproduce faithfully the topology of the original graph including not only the statistics of clustering but also of higher order cycles. Knowing the dimension is important because, in our model, it tells us the number of independent similarity attributes we need to fully describe the real network under study (the total number of attributes per node is the D similarity attributes plus the hidden degree).

As already explained in the introduction of our manuscript, “our approach is model-driven and assumes

that real networks are well described by the geometric soft configuration model in D dimensions, the $\mathbb{S}^D/\mathbb{H}^{D+1}$ model, which is a multidimensional generalization of the \mathbb{S}^1 model [32] and its \mathbb{H}^2 purely geometric formulation in hyperbolic space [33]. The $\mathbb{S}^1/\mathbb{H}^2$ model is based on fundamental principles to describe the observed connectivity of real unweighted and undirected networks. The model assumes a one-dimensional similarity space plus a popularity dimension from which hyperbolic geometry emerges as the geometry that naturally embodies the hierarchical architecture of networks. In this way, the model explains many typical features of real networks including sparsity, the small-world property, heterogeneous degree distributions, and high levels of clustering [34]. The model is able to do that while being a maximum entropy model (see Ref [34]), meaning that it makes the minimum number of assumptions to explain observations given the constraints (degree distribution and level of clustering), so it is the most parsimonious option providing the simplest explanation of the complex topology of real networks.

It is worth stressing that, in addition to providing such simple explanation to the aforementioned topological properties, the model has proved to explain and predict other far more subtle and fine-grained structural features of real complex systems. For instance, in Refs [36], [38], [41], and [51], the angular coordinates of nodes (similarity) were found to be extremely congruent with intrinsic properties of nodes (indeed, in [41] it was even found that the communities detected via their angular coordinates were more congruent with Preferential Trade Agreements than those found through modularity-based community detection). Another striking feature captured by the model (via the mapping to the \mathbb{H}^2 model) is the navigability of real networks; as shown for instance in Refs [36-38], the embeddings can be used as maps to find nearly-shortest paths and route information efficiently. In Ref (Nature Communications 8:14103 (2017)), an extension of the $\mathbb{S}^1/\mathbb{H}^2$ models to weighted networks predicted and reproduced with astounding precision the Triangle Inequality Violation spectrum of real systems, a highly non-trivial relation between weights, degrees, and triangles that no other model known to date can reproduce. Finally, as reported in Ref. [37], the model also predicts the self-similarity of real networks with respect to the geometric renormalization group. Taken all these considerations together, one may dare to say that, while generating networks simultaneously sparse, small-world, highly clustered, and with heterogeneous degree distributions was an original motivation behind this line of research, the power of the resulting models has proved to reach far beyond those properties in numerous occasions.

In the present manuscript, and as we explained in section “Statistics of cycles and their relation with dimensionality”, we work with the \mathbb{S}^D model where each node is assigned a hidden degree, quantifying its popularity or importance, and a position in a similarity space, represented as a D -dimensional sphere. The probability of connection between any pair of nodes takes the form of a gravity law, Eq. (1) in the manuscript, so that more popular and similar nodes are more likely to be connected. The parameter β (or inverse temperature) calibrates the coupling of the network with the underlying metric space and controls the level of clustering in the topology (cycles of length three in the network induced by transitive relations) as a reflection of the triangle inequality in the latent geometry. Finally, the parameter μ controls the average degree of the network. Notice that these are the only parameters in the \mathbb{S}^D model.

As explained in section “Inferring hidden dimensions”, for each real network we created an ensemble of synthetic surrogates using the \mathbb{S}^D model while preserving the original degree distribution (to preserve the consistency of the hidden degrees in the synthetic networks and of the observed degrees in the original graph, we computed hidden degrees using an iterative process that forces a match between expected degrees in the model and observed degrees, see subsection “Estimation of hidden degrees in the “Methods” section) but with different values of the inverse temperature β and dimension D , such that a data-driven classifier is used to infer D from the surrogates that best approximate the real network in terms of not only clustering but also higher order cycles. As we show in our manuscript,

the multidimensional geometric random graph model can reproduce different graph structures, while preserving sparsity and the small-world property, by changing the degree distribution, parameter β that controls clustering, and the dimension D that controls the chordless cycles spectrum of a network. This dependency of the densities in the network ensemble of chordless cycles of different lengths on the dimensionality of the model is the feature that we exploit to implement our dimensionality detection methodology.

Action taken: We have included a brief paragraph in introduction emphasizing the rationale behind our methodology and its model-driven nature.

Comment 2.2: - The fit to obtain D is an average of local quantities, but complex networks may be very heterogeneous, thus this average value may not be representative at all of the graph structure (take a graph with two different regions with different structures, which will make distribution of some of the combinatorial measures bimodal, the mean value). So I believe that applying dimension reduction methods using this dimension as is proposed would not work on such graphs. Thus the use of this dimension for dimensionality reduction methods may be fairly limited to simple graphs.

Reply and action taken: The reviewer is right in the assertion that our methodology does not require global computations. In contrast, it is based on the computation of local quantities and this is, in fact, one of the strengths of our methodology. Again, and in connection to the answer to the previous comment, our methodology does not characterize a real network with a single integer number, we characterize the network using a geometric random graph model, the multidimensional geometric soft configuration model \mathbb{S}^D , that preserves the original degree distribution and such that the dimensionality has the optimal value for reproducing the topology of the original graph, in particular the statistics of chordless cycles. In this way, our methodology is able to reproduce all the pivotal features that are ubiquitous in real complex networks, including sparsity, the small-world property, heterogeneous degree distribution, high level of clustering and, as proven in the present work, also the distribution of squares and pentagons, and so our methodology is, in general, valid. Please, see answer to comment 2.1 for a more extended explanation.

Regarding the graph structures mentioned by the referee with distribution of some of the combinatorial measures bimodal, this is not what it is found at the local level in the chordless cycles spectrum that we use to determine the dimensionality of the real complex networks analyzed in this work. One has to go to the mesoscale of complex networks or to multilayer structures to find the kind of heterogeneity mentioned by the referee, which is out of scope in this work. Notice also that we work with normalized densities of cycles in the range $[0, 1]$ so that heterogeneity in such measures is strongly limited. As an example, in Fig. 1 below we show the probability densities of C_t, C_s, C_p for the EUEmail network. In all cases, these distributions are single peaked, indicating that the averages that we measure in our analysis are meaningful properties of the network.

Figure 1: Probability densities of C_t, C_s, C_p for the EUEmail network.

Comment 2.3: - It is claimed that there is no need for spatial embedding, but the dimension of the hyperbolic space can be seen as an embedding, so this method can be seen as an 'indirect hyperbolic spatial embedding', highly dependent on the choices of the combinatorial measures used to find the best hyperbolic surrogate. It may be that these measures are the most relevant for this task, but it makes the notion of dimension depend on several non-trivial modeling choices, and therefore cannot be shown to be a 'fundamental measure', as one expect from the natural notion of dimension.

Reply: The reviewer is right when they asserts that our goal is to find the best hyperbolic surrogate of a real network in the sense that we determine the dimensionality of our geometric random graph model so that it is the case. We based our methodology in preserving the degree distribution and selecting the dimension of the geometric random graph model that best reproduces the topological statistics of the network in terms of chordless cycles of different lengths. However, what we understand by embedding a network in a latent space, see Ref. [35] in our manuscript, is very different from this. In fact, it corresponds to reversing what we do in this paper, where we use the model as a network generator. To do an embedding is to reverse engineering this action to find the positions of the nodes in the underlying metric space such that model networks based on the inferred coordinates give the maximum likelihood to obtain the observed structure in the real network. Multidimensional embedding methods are beyond the scope of the present manuscript. Let us just add here that we are already working on such a methodology to embed real networks into the \mathbb{S}^D multidimensional hyperbolic space and it is far from trivial to find the efficient and computationally feasible algorithm.

In our geometric random graph model, the notion of dimension does not depend on modeling choices as suggested by the reviewer. In the \mathbb{S}^D model, a node is assigned a hidden degree, quantifying its popularity or importance, and a position in a similarity space, represented as a D-dimensional sphere. Hence, the dimension refers to the number of coordinates that one needs to specify a point in similarity space. Therefore, in our geometric random graph model the dimension is a fundamental measure. What we prove in this work is that this dimension is related to the chordless cycles spectrum in the network ensemble.

Comment 2.4: - I am not familiar enough with generating these hyperbolic surrogates, but I don't believe it is such an efficient strategy to generate many models, and find the closest match to estimate a single integer number, as is claimed.

Reply: We see that the problem is again that we were not fully successful at explaining clearly our strategy. It is not about estimating an integer, it is about estimating the dimensionality of the underlying geometric random graph model. Please, see answer to comment 2.1 above.

Comment 2.5: - There is no study of comparing this dimension measure with previous ones, or other global graphs quantities (radius, mass, etc...) to show that D provides a non-trivial measure that cannot be obtained via other much simpler computations.

Reply: The dimension that we estimate can only be compared with the dimension obtained by applying other dimensional reduction techniques. It is meaningless to compare it with other global quantities that have nothing to do with dimension. For instance, we are sure that the reviewer would agree with us that it would be meaningless to compare the average shortest path length of a network with its overall level of clustering. In the same way, the dimension that we obtain cannot be compared with other quantities that are not a dimension of a geometric representation of the network. In this respect, we have already compared our estimation with other alternatives, typically based on Euclidean space, see for instance Refs. 8,12, and 57 in the numbering of the new version of the manuscript. Even if, in all of these references, the detected dimensionality is much lower than the number of nodes, it is still much above our estimations because Euclidean space is not the most appropriate geometry to represent

the hierarchical structure of real complex networks. As we prove in our manuscript, hyperbolic space offers thus an advantage when compared to Euclidean space and it is very well suited to embed real world networks and to produce ultra-low dimensional representations.

As pointed out by the first reviewer in their report, “The findings from the analysis of real-world networks are particularly impressive especially in view of results reported in recent literature. It seems that hyperbolic space is very well suited to embed (some) real world networks compared to Euclidean space. To make a concrete example, the AS-level Internet requires $D=8$ dimensions to be embedded in hyperbolic space according to the present analysis, but $D=47$ in Euclidean space [see for example Phys. Rev. E 103, 012305 (2021)].” (Ref. 57 in the new version)

Action taken: We have extended the information in the text about how our estimations compare with alternative estimations. We have also included a new citation to an alternative method. Notice that after the changes in the new version of the manuscript, we found that the AS-level Internet requires $D=7$ dimensions to be embedded in hyperbolic space.

Comment 2.6: - There is no analysis of what the dimension of simple geometric graphs would be, such as a grid/meshes. Would we get $D=2$ for a simple grid graph, and $D=1$ for a line graph? I would expect a meaningful measure of dimension would satisfy this basic concept.

Reply: Apart from complex networks, our formalism is also valid for geometric graphs as far as they are random, which are very homogeneous and clustered and non small world networks, which happens in our model when $\gamma > 3$ and $\beta > 2D$. This means that graphs in this region of the parameter space are effectively described by the geometry of the D -sphere, which becomes the Euclidean space in the thermodynamic limit, rather than by hyperbolic geometry.

However, D -dimensional lattices are clearly out of the category of complex networks and of geometric random graphs, the position of their nodes in the space is highly ordered and their links are strongly correlated whereas in our model links are statistical independent. D -dimensional lattices are, thus, not suited to being described by our hyperbolic multidimensional model.

A different question is that of real networks naturally embedded in Euclidean space. If those present the typical features of being small worlds with high clustering, like brain connectomes, or they are geometric random graphs, then our methodology is definitely applicable. This means that explicit distance in such complex networks naturally embedded in Euclidean space may not be the only factor that determines connectivity in the network, and other features may have a role as well.

Action taken: We have included in section “Discussion” a paragraph to clarify the limitations of the proposed methodology where we also discuss the situation of networks with explicit geometry.

Comment 2.7: Overall, I don't think this paper is suited for Nature Communication, but rather for a more specialized network science journal, provided some of the points above are addressed.

Reply: We thank the reviewer for their time reviewing our paper and for the valuable comments, which we believe have helped us improve the manuscript's readability substantially. We are sorry that we were not fully successful at explaining clearly the model and its importance. We concede that our models and methods are sophisticated. This is partly due to the fact that our geometric approach to complex networks has been developed during more than a decade, resulting in a new branch of research in network science that is now called network geometry. In the new version of the manuscript, we have made a serious effort to present more clearly our results. We hope that the new version of the manuscript, extensively revised according to the comments, is better and more intelligible and will

help any reader to discern more clearly the importance of our work.

Let us say here that we are convinced that our results are important for a general audience beyond the set of true experts on the type of models discussed in our work. As we already pointed out in the introduction, in this work we propose a methodology to detect network dimensionality that is based on models in hyperbolic geometry, which is the natural geometry to embed real world complex networks as we extensively proved in a series of previous works, see *Nature Reviews Physics* 3, 114–135 (2021) and *The Shortest Path to Network Geometry: A Practical Guide to Basic Models and Applications (Elements in Structure and Dynamics of Complex Networks)*. Cambridge: Cambridge University Press (2022), and references therein. The dimensionality detection technique presents strong advantages. First, it does not require to actually perform the embedding of a network in order to estimate its optimal embedding dimension. Second, our methodology does not require global computations but uses the spectrum of short cycles. And third, the findings from the analysis of real-world networks are particularly impressive especially in view of results reported in recent literature. To give a concrete example, and as mentioned by the first reviewer in their report, the AS-level Internet requires $D = 7$ dimensions to be embedded in hyperbolic space according to the present analysis, but $D = 47$ in Euclidean space or even more [see for example *Nature Communications* 12(1),1–10 (2021) and *Physical Review E* 103, 012305 (2021)]. Given that networks are ubiquitous across disciplines, we believe that our results are of interest for a broad spectrum of scientists in different fields.

REVIEWERS' COMMENTS

Reviewer #1 (Remarks to the Author):

I thank the authors for the careful revisions, and the detailed replies that they provided. I am happy to know that some of my comments were useful to improve the quality of this, already excellent, work. I strongly support the publication of the present manuscript without any further delay.

Reviewer #2 (Remarks to the Author):

Thank you for the more detailed explanations on this work on your previous works on this topic of hyperbolic geometry in complex networks, it indeed shed more light on the presented results.

I have never been myself very convinced that the topic of embedding networks is interesting as such, as for me, there is no way to prove that this or that embedding is better than another, etc... So my original read of the paper as a tool to guess what could be a potentially good embedding dimension sounded very limited to me. I am also not very convinced with methods trying to fit any generative random network models to a wide range of real-world networks, as none will be accurate by definition -- real world networks have been generated by very complex processes, all very different (an airport network and bain connectome have nothing to do with each others in the way they have 'emerged').

But indeed, as the method itself seems to have some potential for further works on network theory, and reviewer 1, who seemed more expert in this topic than me, is positive, I will also be positive for accepting the paper.

Second resubmission of manuscript NCOMMS-21-42385 Detecting the ultra low dimensionality of real networks

Pedro Almagro, Marián Boguñá, M. Ángeles Serrano

Reviewer #1

Comment 1.0.: I thank the authors for the careful revisions, and the detailed replies that they provided. I am happy to know that some of my comments were useful to improve the quality of this, already excellent, work. I strongly support the publication of the present manuscript without any further delay.

Reply: We thank again the referee for all their constructive comments and suggestions that have certainly helped us to improve our work.

Reviewer #2

Comment 2.0: Thank you for the more detailed explanations on this work on your previous works on this topic of hyperbolic geometry in complex networks, it indeed shed more light on the presented results.

I have never been myself very convinced that the topic of embedding networks is interesting as such, as for me, there is no way to prove that this or that embedding is better than another, etc... So my original read of the paper as a tool to guess what could be a potentially good embedding dimension sounded very limited to me. I am also not very convinced with methods trying to fit any generative random network models to a wide range of real-world networks, as none will be accurate by definition – real world networks have been generated by very complex processes, all very different (an airport network and bain connectome have nothing to do with each others in the way they have 'emerged').

But indeed, as the method itself seems to have some potential for further works on network theory, and reviewer 1, who seemed more expert in this topic than me, is positive, I will also be positive for accepting the paper.

Reply: We thank the referee for their comments, which have helped us to improve the presentation of our work. We agree with the referee in that real networks from different domains might emerge from very different mechanisms. However, the most fundamental contribution of network science is the discovery that there is a set of topological features that are common to the vast majority of real networks, even if they are from completely different domains. These are the sparsity of the number of connections, the small-world property, the high levels of clustering, the heterogeneity of the degree distribution, the existence of communities, etc. The ubiquity of these properties suggests that the different processes at play in the emergence of real networks may lead to an effective connectivity law that can be applied to networks from different domains. The power of our geometric model is that with very few parameters and with a very simple connectivity law, it is able to recover all these properties. From this point of view, our model is the most parsimonious explanation for the observed topologies of many real networks.